# Sample-Efficient Deep Reinforcement Learning via Episodic Backward Update

**Su Young Lee,    Sungik Choi,    Sae-Young Chung**
School of Electrical Engineering, KAIST, Republic of Korea
{suyoung.l, si_choi, schung}@kaist.ac.kr

## Abstract

We propose Episodic Backward Update (EBU) – a novel deep reinforcement learning algorithm with a direct value propagation. In contrast to the conventional use of the experience replay with uniform random sampling, our agent samples a whole episode and successively propagates the value of a state to its previous states. Our computationally efficient recursive algorithm allows sparse and delayed rewards to propagate directly through all transitions of the sampled episode. We theoretically prove the convergence of the EBU method and experimentally demonstrate its performance in both deterministic and stochastic environments. Especially in 49 games of Atari 2600 domain, EBU achieves the same mean and median human normalized performance of DQN by using only 5% and 10% of samples, respectively.

## 1 Introduction

Deep reinforcement learning (DRL) has been successful in many complex environments such as the Arcade Learning Environment [2] and Go [18]. Despite DRL's impressive achievements, it is still impractical in terms of sample efficiency. To achieve human-level performance in the Arcade Learning Environment, Deep $Q$-Network (DQN) [14] requires 200 million frames of experience for training which corresponds to 39 days of gameplay in real-time. Clearly, there is still a tremendous gap between the learning process of humans and that of deep reinforcement learning agents. This problem is even more crucial for tasks such as autonomous driving, where we cannot risk many trials and errors due to the high cost of samples.

One of the reasons why DQN suffers from such low sample efficiency is the sampling method from the replay memory. In many practical problems, an RL agent observes sparse and delayed rewards. There are two main problems when we sample one-step transitions uniformly at random. **(1)** We have a low chance of sampling a transition with a reward for its sparsity. The transitions with rewards should always be updated to assign credits for actions that maximize the expected return. **(2)** In the early stages of training when all values are initialized to zero, there is no point in updating values of one-step transitions with zero rewards if the values of future transitions with nonzero rewards have not been updated yet. Without the future reward signals propagated, the sampled transition will always be trained to return a zero value.

In this work, we propose Episodic Backward Update (EBU) to present solutions for the problems raised above. When we observe an event, we scan through our memory and seek for the past event that caused the later one. Such an episodic control method is how humans normally recognize the cause and effect relationship [10]. Inspired by this, we can solve the first problem **(1)** by sampling transitions in an episodic manner. Then, we can be assured that at least one transition with a non-zero reward is used for the value update. We can solve the second problem **(2)** by updating the values of transitions in a backward manner in which the transitions were made. Afterward, we can perform an

efficient reward propagation without any meaningless updates. This method faithfully follows the principle of dynamic programming.

As mentioned by the authors of DQN, updating correlated samples in a sequence is vulnerable to overestimation. In Section 3, we deal with this issue by adopting a diffusion factor to mediate between the learned values from the future transitions and the current sample reward. In Section 4, we theoretically prove the convergence of our method for both deterministic and stochastic MDPs. In Section 5, we empirically show the superiority of our method on 2D MNIST Maze Environment and the 49 games of Atari 2600 domain. Especially in 49 games of the Atari 2600 domain, our method requires only 10M frames to achieve the same mean human-normalized score reported in Nature DQN [14], and 20M frames to achieve the same median human-normalized score. Remarkably, EBU achieves such improvements with a comparable amount of computation complexity by only modifying the target generation procedure for the value update from the original DQN.

## 2   Background

The goal of reinforcement learning (RL) is to learn the optimal policy that maximizes the expected sum of rewards in the environment that is often modeled as a Markov decision process (MDP) $M = (\mathcal{S}, \mathcal{A}, P, R)$. $\mathcal{S}$ denotes the state space, $\mathcal{A}$ denotes the action space, $P : \mathcal{S} \times \mathcal{A} \times \mathcal{S} \rightarrow \mathbb{R}$ denotes the transition probability distribution, and $R : \mathcal{S} \times \mathcal{A} \rightarrow \mathbb{R}$ denotes the reward function. $Q$-learning [22] is one of the most widely used methods to solve RL tasks. The objective of $Q$-learning is to estimate the state-action value function $Q(s, a)$, or the $Q$-function, which is characterized by the Bellman optimality equation. $Q^*(s_t, a) = \mathbb{E}[r_t + \gamma \max_{a'} Q^*(s_{t+1}, a')]$.

There are two major inefficiencies of the traditional on-line $Q$-learning. First, each experience is used only once to update the $Q$-function. Secondly, learning from experiences in a chronologically forward order is much more inefficient than learning in a chronologically backward order, because the value of $s_{t+1}$ is required to update the value of $s_t$. Experience replay [12] is proposed to overcome these inefficiencies. After observing a transition $(s_t, a_t, r_t, s_{t+1})$, the agent stores the transition into its replay buffer. In order to learn the $Q$-values, the agent samples transitions from the replay buffer.

In practice, the state space $\mathcal{S}$ is extremely large, therefore it is impractical to tabularize the $Q$-values of all state-action pairs. Deep $Q$-Network [14] overcomes this issue by using deep neural networks to approximate the $Q$-function. DQN adopts experience replay to use each transition for multiple updates. Since DQN uses a function approximator, consecutive states output similar $Q$-values. If DQN updates transitions in a chronologically backward order, often overestimation errors cumulate and degrade the performance. Therefore, DQN does not sample transitions in a backward order, but uniformly at random. This process breaks down the correlations between consecutive transitions and reduces the variance of updates.

There have been a variety of methods proposed to improve the performance of DQN in terms of stability, sample efficiency, and runtime. Some methods propose new network architectures. The dueling network architecture [21] contains two streams of separate $Q$-networks to estimate the value functions and the advantage functions. Neural episodic control [16] and model-free episodic control [5] use episodic memory modules to estimate the state-action values. RUDDER [1] introduces an LSTM network with contribution analysis for an efficient return decomposition. Ephemeral Value Adjustments (EVA) [7] combines the values of two separate networks, where one is the standard DQN and another is a trajectory-based value network.

Some methods tackle the uniform random sampling replay strategy of DQN. Prioritized experience replay [17] assigns non-uniform probability to sample transitions, where greater probability is assigned for transitions with higher temporal difference (TD) error. Inspired by Lin's backward use of replay memory, some methods try to aggregate TD values with Monte-Carlo (MC) returns. $Q(\lambda)$ [23], $Q^*(\lambda)$ [6] and Retrace($\lambda$) [15] modify the target values to allow the on-policy samples to be used interchangeably for on-policy and off-policy learning. Count-based exploration method combined with intrinsic motivation [3] takes a mixture of one-step return and MC return to set up the target value. Optimality Tightening [8] applies constraints on the target using the values of several neighboring transitions. Simply by adding a few penalty terms to the loss, it efficiently propagates reliable values to achieve fast convergence.

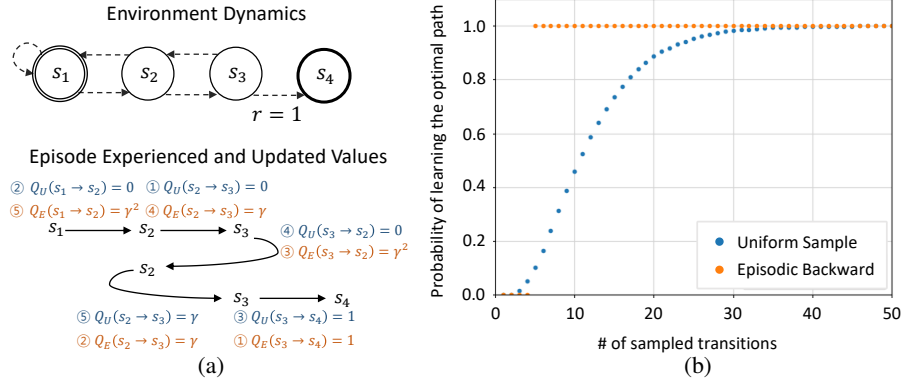

Figure 1: A motivating example where uniform sampling method fails but EBU does not. (a): A simple navigation domain with 4 states and a single rewarded transition. Circled numbers indicate the order of sample updates. $Q_U$ and $Q_E$ stand for the $Q$-values learned by the uniform random sampling method and the EBU method respectively. (b): The probability of learning the optimal path $(s_1 \to s_2 \to s_3 \to s_4)$ after updating the $Q$-values with sample transitions.

Our goal is to improve the sample efficiency of deep reinforcement learning by making a simple yet effective modification. Without a single change of the network structure, training schemes, and hyperparameters of the original DQN, we only modify the target generation method. Instead of using a limited number of transitions, our method samples a whole episode from the replay memory and propagates the values sequentially throughout the entire transitions of the sampled episode in a backward manner. By using a temporary backward $Q$-table with a diffusion coefficient, our novel algorithm effectively reduces the errors generated from the consecutive updates of correlated states.

## 3 Proposed Methods

### 3.1 Episodic Backward Update for Tabular $Q$-Learning

Let us imagine a simple tabular MDP with a single rewarded transition (Figure 1, (a)), where an agent can only take one of the two actions: *'left'* and *'right'*. In this example, $s_1$ is the initial state, and $s_4$ is the terminal state. A reward of 1 is gained only when the agent reaches the terminal state and a reward of 0 is gained from any other transitions. To make it simple, assume that we have only one episode stored in the experience memory: $(s_1 \to s_2 \to s_3 \to s_2 \to s_3 \to s_4)$. The $Q$-values of all transitions are initialized to zero. With a discount $\gamma \in (0, 1)$, the optimal policy is to take the action *'right'* in all states. When sampling transitions uniformly at random as Nature DQN, the key transitions $(s_1 \to s_2)$, $(s_2 \to s_3)$ and $(s_3 \to s_4)$ may not be sampled for updates. Even when those transitions are sampled, there is no guarantee that the update of the transition $(s_3 \to s_4)$ is done before the update of $(s_2 \to s_3)$. We can speed up the reward propagation by updating all transitions within the episode in a backward manner. Such a recursive update is also computationally efficient.

We can calculate the probability of learning the optimal path $(s_1 \to s_2 \to s_3 \to s_4)$ as a function of the number of sample transitions trained. With the tabular Episodic Backward Update stated in Algorithm 1, which is a special case of Lin's algorithm [11] with recency parameter $\lambda = 0$, the agent can figure out the optimal policy just after 5 updates of $Q$-values. However, we see that the uniform sampling method requires more than 40 transitions to learn the optimal path with probability close to 1 (Figure 1, (b)).

Note that this method differs from the standard $n$-step $Q$-learning [22]. In $n$-step $Q$-learning, the number of future steps for the target generation is fixed as $n$. However, our method considers $T$ future values, where $T$ is the length of the sampled episode. $N$-step $Q$-learning takes a max operator at the $n$-th step only, whereas our method takes a max operator at every iterative backward step which can propagate high values faster. To avoid exponential decay of the $Q$-value, we set the learning rate $\alpha = 1$ within the single episode update.

---

**Algorithm 1** Episodic Backward Update for Tabular $Q$-Learning (single episode, tabular)

---

1: Initialize the $Q$- table $Q \in \mathbb{R}^{\mathcal{S} \times \mathcal{A}}$ with all-zero matrix.
   $Q(s, a) = 0$ for all state action pairs $(s, a) \in \mathcal{S} \times \mathcal{A}$.
2: Experience an episode $E = \{(s_1, a_1, r_1, s_2), \ldots, (s_T, a_T, r_T, s_{T+1})\}$
3: **for** $t = T$ to 1 **do**
4:    $Q(s_t, a_t) \leftarrow r_t + \gamma \max_{a'} Q(s_{t+1}, a')$
5: **end for**

---

---

**Algorithm 2** Episodic Backward Update

---

1: **Initialize**: replay memory $D$ to capacity $N$, on-line action-value function $Q(\cdot; \boldsymbol{\theta})$, target action-value function $\hat{Q}(\cdot; \boldsymbol{\theta}^-)$
2: **for** episode = 1 to $M$ **do**
3:    **for** $t = 1$ to Terminal **do**
4:        With probability $\epsilon$ select a random action $a_t$, otherwise select $a_t = \operatorname{argmax}_a Q(s_t, a; \boldsymbol{\theta})$
5:        Execute action $a_t$, observe reward $r_t$ and next state $s_{t+1}$
6:        Store transition $(s_t, a_t, r_t, s_{t+1})$ in $D$
7:        Sample a random episode $E = \{\boldsymbol{S}, \boldsymbol{A}, \boldsymbol{R}, \boldsymbol{S'}\}$ from $D$, set $T = \text{length}(E)$
8:        Generate a temporary target $Q$-table, $\tilde{Q} = \hat{Q}(\boldsymbol{S'}, \cdot; \boldsymbol{\theta}^-)$
9:        Initialize the target vector $\boldsymbol{y} = \text{zeros}(T)$, $\boldsymbol{y}_T \leftarrow \boldsymbol{R}_T$
10:       **for** $k = T - 1$ to 1 **do**
11:           $\tilde{Q}[\boldsymbol{A}_{k+1}, k] \leftarrow \beta \boldsymbol{y}_{k+1} + (1 - \beta)\tilde{Q}[\boldsymbol{A}_{k+1}, k]$
12:           $\boldsymbol{y}_k \leftarrow \boldsymbol{R}_k + \gamma \max_a \tilde{Q}[a, k]$
13:       **end for**
14:       Perform a gradient descent step on $(\boldsymbol{y} - Q(\boldsymbol{S}, \boldsymbol{A}; \boldsymbol{\theta}))^2$ with respect to $\boldsymbol{\theta}$
15:       Every $C$ steps reset $\hat{Q} = Q$
16:   **end for**
17: **end for**

---

There are some other multi-step methods that converge to the optimal state-action value function, such as $Q(\lambda)$ and $Q^*(\lambda)$. However, our algorithm neither cuts trace of trajectories as $Q(\lambda)$, nor requires the parameter $\lambda$ to be small enough to guarantee convergence as $Q^*(\lambda)$. We present a detailed discussion on the relationship between EBU and other multi-step methods in Appendix F.

### 3.2    Episodic Backward Update for Deep $Q$-Learning[1]

Directly applying the backward update algorithm to deep reinforcement learning is known to show highly unstable results due to the high correlation of consecutive samples. We show that the fundamental ideas of the tabular version of the backward update algorithm may be applied to its deep version with just a few modifications. The full algorithm introduced in Algorithm 2 closely resembles that of Nature DQN [14]. Our contributions lie in the recursive backward target generation with a diffusion factor $\beta$ (starting from line number 7 of Algorithm 2), which prevents the overestimation errors from correlated states cumulating.

Instead of sampling transitions uniformly at random, we make use of all transitions within the sampled episode $E = \{\boldsymbol{S}, \boldsymbol{A}, \boldsymbol{R}, \boldsymbol{S'}\}$. Let the sampled episode start with a state $S_1$, and contain T transitions. Then $E$ can be denoted as a set of four length-$T$ vectors: $\boldsymbol{S} = \{S_1, S_2, \ldots, S_T\}$; $\boldsymbol{A} = \{A_1, A_2, \ldots, A_T\}$; $\boldsymbol{R} = \{R_1, R_2, \ldots, R_T\}$ and $\boldsymbol{S'} = \{S_2, S_3, \ldots, S_{T+1}\}$. The temporary target $Q$-table, $\tilde{Q}$ is an $|\mathcal{A}| \times T$ matrix which stores the target $Q$-values of all states $\boldsymbol{S'}$ for all valid actions, where $\mathcal{A}$ is the action space of the MDP. Therefore, the $j$-th column of $\tilde{Q}$ is a column vector that contains $\hat{Q}(S_{j+1}, a; \boldsymbol{\theta}^-)$ for all valid actions $a \in \mathcal{A}$, where $\hat{Q}$ is the target $Q$-function parametrized by $\boldsymbol{\theta}^-$.

After the initialization of the temporary $Q$-table, we perform a recursive backward update. Adopting the backward update idea, one element $\tilde{Q}[\boldsymbol{A}_{k+1}, k]$ in the $k$-th column of the $\tilde{Q}$ is replaced using the next transition's target $\boldsymbol{y}_{k+1}$. Then $\boldsymbol{y}_k$ is estimated as the maximum value of the newly modified $k$-th column of $\tilde{Q}$. Repeating this procedure in a recursive manner until the start of the episode, we can

successfully apply the backward update algorithm for a deep $Q$-network. The process is described in detail with a supplementary diagram in Appendix E.

We are using a function approximator, and updating correlated states in a sequence. As a result, we observe overestimated values propagating and compounding through the recursive $\max$ operations. We solve this problem by introducing the diffusion factor $\beta$. By setting $\beta \in (0, 1)$, we can take a weighted sum of the new backpropagated value and the pre-existing value estimate. One can regard $\beta$ as a learning rate for the temporary $Q$-table, or as a level of *'backwardness'* of the update. This process stabilizes the learning process by exponentially decreasing the overestimation error. Note that Algorithm 2 with $\beta = 1$ is identical to the tabular backward algorithm stated in Algorithm 1. When $\beta = 0$, the algorithm is identical to episodic one-step DQN. The role of $\beta$ is investigated in detail with experiments in Section 5.3.

### 3.3 Adaptive Episodic Backward Update for Deep $Q$-Learning

The optimal diffusion factor $\beta$ varies depending on the type of the environment and the degree of how much the network is trained. We may further improve EBU by developing an adaptive tuning scheme for $\beta$. Without increasing the sample complexity, we propose an adaptive, single actor and multiple learner version of EBU. We generate $K$ learner networks with different diffusion factors, and a single actor to output a policy. For each episode, the single actor selects one of the learner networks in a regular sequence. Each learner is trained in parallel, using the same episode sampled from a shared experience replay. Even with the same training data, all learners show different interpretations of the sample based on the different levels of trust in backwardly propagated values. We record the episode scores of each learner during training. After every fixed step, we synchronize all the learner networks with the parameters of a learner network with the best training score. This adaptive version of EBU is presented as a pseudo-code in Appendix A. In Section 5.2, we compare the two versions of EBU, one with a constant $\beta$ and another with an adaptive $\beta$.

## 4 Theoretical Convergence

### 4.1 Deterministic MDPs

We prove that Episodic Backward Update with $\beta \in (0, 1)$ defines a contraction operator, and converges to the optimal $Q$-function in finite and deterministic MDPs.

**Theorem 1.** *Given a finite, deterministic and tabular MDP $M = (\mathcal{S}, \mathcal{A}, P, R)$, the Episodic Backward Update algorithm in Algorithm 2 converges to the optimal Q-function w.p. 1 as long as*

• *The step size satisfies the Robbins-Monro condition;*

• *The sample trajectories are finite in lengths $l$: $\mathbb{E}[l] < \infty$;*

• *Every (state, action) pair is visited infinitely often.*

We state the proof of Theorem 1 in Appendix G. Furthermore, even in stochastic environments, we can guarantee the convergence of the episodic backward algorithm for a sufficiently small $\beta$.

### 4.2 Stochastic MDPs

**Theorem 2.** *Given a finite, tabular and stochastic MDP $M = (\mathcal{S}, \mathcal{A}, P, R)$, define $R_{\max}^{\text{sto}}(s, a)$ as the maximal return of trajectory that starts from state $s \in \mathcal{S}$ and action $a \in \mathcal{A}$. In a similar way, define $r_{\min}^{\text{sto}}(s, a)$ and $r_{\text{mean}}^{\text{sto}}(s, a)$ as the minimum and mean of possible reward by selecting action $a$ in state $s$. Define $\mathcal{A}_{\text{sub}}(s) = \{a' \in \mathcal{A} | Q^*(s, a') < \max_{a \in \mathcal{A}} Q^*(s, a)\}$ as the set of suboptimal actions in state $s \in \mathcal{S}$. Define $\mathcal{A}_{\text{opt}}(s) = \mathcal{A} \backslash \mathcal{A}_{\text{sub}}(s)$. Then, under the conditions of Theorem 1, and*

$$\beta \leq \inf_{s \in \mathcal{S}} \inf_{a' \in \mathcal{A}_{\text{sub}}(s)} \inf_{a \in \mathcal{A}_{\text{opt}}(s)} \frac{Q^*(s, a) - Q^*(s, a')}{R_{\max}^{\text{sto}}(s, a') - Q^*(s, a')}, \tag{1}$$

$$\beta \leq \inf_{s \in \mathcal{S}} \inf_{a' \in \mathcal{A}_{\text{sub}}(s)} \inf_{a \in \mathcal{A}_{\text{opt}}(s)} \frac{Q^*(s, a) - Q^*(s, a')}{r_{\text{mean}}^{\text{sto}}(s, a) - r_{\min}^{\text{sto}}(s, a)}, \tag{2}$$

*the Episodic Backward Update algorithm in Algorithm 2 converges to the optimal Q-function w.p. 1.*

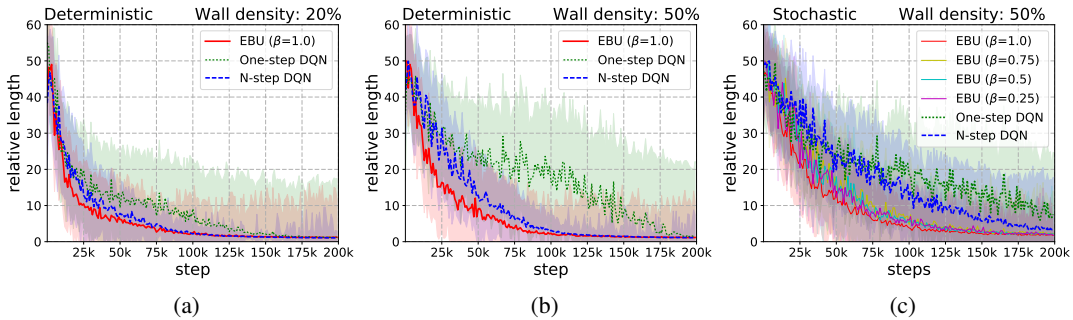

(a)                                              (b)                                              (c)

Figure 2: (a) & (b): Median of 50 relative lengths of EBU and baselines. EBU outperforms other baselines significantly in the low sample regime and for high wall density. (c): Median relative lengths of EBU and other baseline algorithms in MNIST maze with stochastic transitions.

The main intuition of this theorem is that $\beta$ acts as a learning rate of the backward target therefore mitigates the collision between the max operator and stochastic transitions.

## 5 Experimental Results

### 5.1 2D MNIST Maze (Deterministic/Stochastic MDPs)

We test our algorithm in the 2D Maze Environment. Starting from the initial position at $(0,0)$, the agent has to navigate through the maze to reach the goal position at $(9,9)$. To minimize the correlation between neighboring states, we use the MNIST dataset [9] for the state representation. The agent

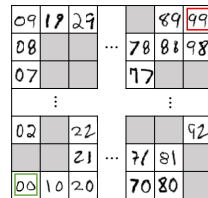

Figure 3: 2D MNIST Maze

receives the coordinates of the position in two MNIST images as the state representation. The training environments are 10 by 10 mazes with randomly placed walls. We assign a reward of 1000 for reaching the goal, and a reward of -1 for bumping into a wall. A wall density indicates the probability of having a wall at each position. For each wall density, we generate 50 random mazes with different wall locations. We train a total of 50 independent agents, one for each maze over 200,000 steps. The performance metric, relative length is defined as $l_{\mathrm{rel}} = l_{\mathrm{agent}}/l_{\mathrm{oracle}}$, which is the ratio between the length of the agent's path $l_{\mathrm{agent}}$ and the length of the ground truth shortest path $l_{\mathrm{oracle}}$ to reach the goal. The details of the hyperparameters and the network structure are described in Appendix D.

We compare EBU to uniform random sampling one-step DQN and $n$-step DQN. For $n$-step DQN, we set the value of $n$ as the length of the episode. Since all three algorithms eventually achieve median relative lengths of 1 at the end of the training, we report the relative lengths at 100,000 steps in Table 1. One-step DQN performs the worst in all configurations, implying the inefficiency of uniform sampling update in environments with sparse and delayed rewards. As the wall density increases, it becomes more important for the agent to learn the correct decisions at bottleneck positions. $N$-step DQN shows the best performance with a low wall density, but as the wall density increases, EBU significantly outperforms $n$-step DQN.

In addition, we run experiments with stochastic transitions. We assign 10% probability for each side action for all four valid actions. For example, when an agent takes an action '*up*', there is a 10% chance of transiting to the left state, and 10% chance of transiting to the right state. In Figure 2 (c), we see that the EBU agent outperforms the baselines in the stochastic environment as well.

Table 1: Relative lengths (Mean & Median) of 50 deterministic MNIST Maze after 100,000 steps

| Wall density | EBU ($\beta = 1.0$) | | One-step DQN | | $N$-step DQN | |
|---|---|---|---|---|---|---|
| 20% | 5.44 | 2.42 | 14.40 | 9.25 | **3.26** | **2.24** |
| 30% | **8.14** | **3.03** | 25.63 | 21.03 | 8.88 | 3.32 |
| 40% | **8.61** | **2.52** | 25.45 | 22.71 | 8.96 | 3.50 |
| 50% | **5.51** | **2.34** | 22.36 | 16.62 | 11.32 | 3.12 |

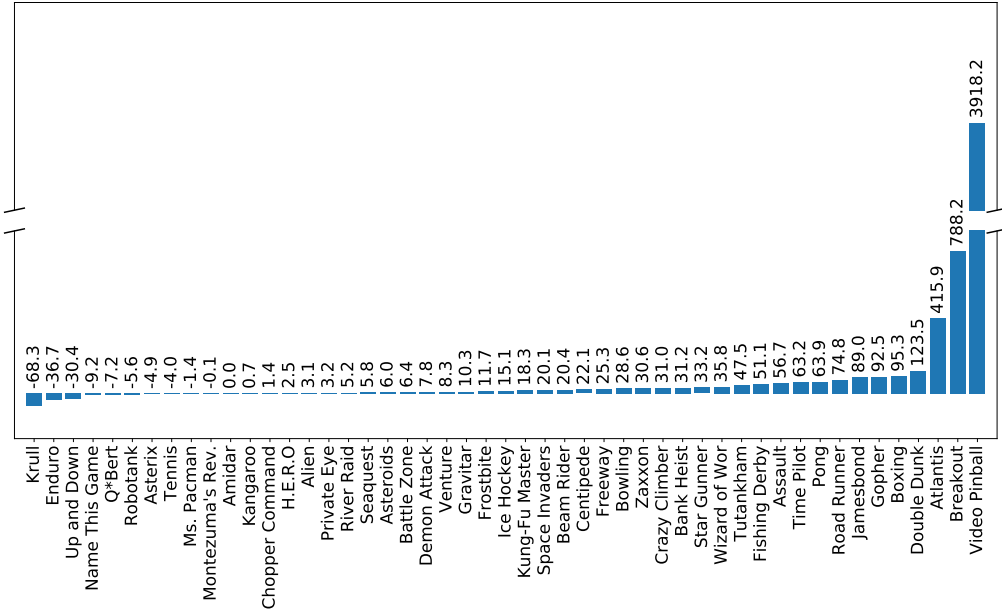

Figure 4: Relative score of adaptive EBU (4 random seeds) compared to Nature DQN (8 random seeds) in percents (%) both trained for 10M frames.

## 5.2 49 Games of Atari 2600 Environment (Deterministic MDPs)

The Arcade Learning Environment [2] is one of the most popular RL benchmarks for its diverse set of challenging tasks. We use the same set of 49 Atari 2600 games, which was evaluated in Nature DQN paper [14].

We select $\beta = 0.5$ for EBU with a constant diffusion factor. For adaptive EBU, we train $K = 11$ parallel learners with diffusion factors $0.0, 0.1, \ldots$, and $1.0$. We synchronize the learners at the end of each epoch (0.25M frames). We compare our algorithm to four baselines: Nature DQN [14], Prioritized Experience Replay (PER) [17], Retrace($\lambda$) [15] and Optimality Tightening (OT) [8]. We train EBU and baselines for 10M frames (additional 20M frames for adaptive EBU) on 49 Atari games with the same network structure, hyperparameters, and evaluation methods used in Nature DQN. The choice of such a small number of training steps is made to investigate the sample efficiency of each algorithm following [16, 8]. We report the mean result from 4 random seeds for adaptive EBU and 8 random seeds for all other baselines. Detailed specifications for each baseline are described in Appendix D.

First, we show the improvement of adaptive EBU over Nature DQN at 10M frames for all 49 games in Figure 4. To compare the performance of an agent to its baseline's, we use the following relative score, $\frac{\text{Score}_{\text{Agent}} - \text{Score}_{\text{Baseline}}}{\max\{\text{Score}_{\text{Human}}, \text{Score}_{\text{Baseline}}\} - \text{Score}_{\text{Random}}}$ [21]. This measure shows how well an agent performs a task compared to the task's level of difficulty. EBU ($\beta = 0.5$) and adaptive EBU outperform Nature DQN in 33 and 39 games out of 49 games, respectively. The large amount of improvements in games such as "Atlantis," "Breakout," and "Video Pinball" highly surpass minor failings in few games.

We use human-normalized score, $\frac{\text{Score}_{\text{Agent}} - \text{Score}_{\text{Random}}}{|\text{Score}_{\text{Human}} - \text{Score}_{\text{Random}}|}$ [20], which is the most widely used metric to make an apple-to-apple comparison in the Atari domain. We report the mean and the median human-normalized scores of the 49 games in Table 2. The result signifies that our algorithm outperforms the baselines in both the mean and median of the human-normalized scores. PER and Retrace($\lambda$) do not show a lot of improvements for a small number of training steps as 10M frames. Since OT has to calculate the $Q$-values of neighboring states and compare them to generate the penalty term, it requires about 3 times more training time than Nature DQN. However, EBU performs iterative episodic updates using the temporary $Q$-table that is shared by all transitions in the episode, EBU has almost the same computational cost as that of Nature DQN.

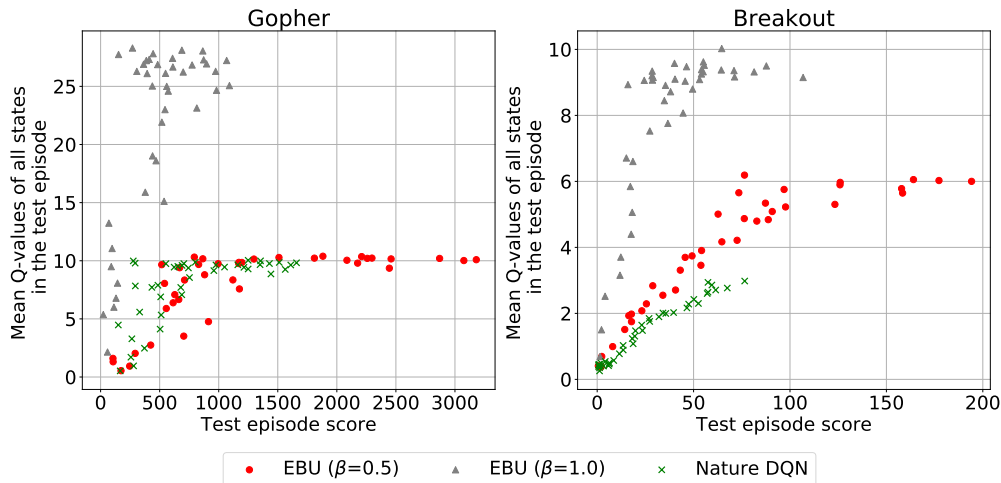

Figure 5: Episode scores and average $Q$-values of all state-action pairs in "Gopher" and "Breakout".

The most significant result is that EBU ($\beta = 0.5$) requires only 10M frames of training to achieve the mean human-normalized score reported in Nature DQN, which is trained for 200M frames. Although 10M frames are not enough to achieve the same median score, adaptive EBU trained for 20M frames achieves the median normalized score. These results signify the efficacy of backward value propagation in the early stages of training. Raw scores for all 49 games are summarized in Appendix B. Learning curves of adaptive EBU for all 49 games are reported in Appendix C.

Table 2: Summary of training time and human-normalized performance. Training time refers to the total time required to train 49 games of 10M frames using a single NVIDIA TITAN Xp for a single random seed. We use multi-GPUs to train learners of adaptive EBU in parallel. (*) The result of OT differs from the result reported in [8] due to different evaluation methods (i.e. not limiting the maximum number of steps for a test episode and taking maximum score from random seeds). (**) We report the scores of Nature DQN (200M) from [14].

| Algorithm (frames) | Training Time (hours) | Mean (%) | Median (%) |
|---|---|---|---|
| EBU ($\beta = 0.5$) (10M) | 152 | 253.55 | 51.55 |
| EBU (adaptive $\beta$) (10M) | 203 | 275.78 | 63.80 |
| Nature DQN (10M) | 138 | 133.95 | 40.42 |
| PER (10M) | 146 | 156.57 | 40.86 |
| Retrace($\lambda$) (10M) | 154 | 93.77 | 41.99 |
| OT (10M)* | 407 | 162.66 | 49.42 |
| EBU (adaptive $\beta$) (20M) | 450 | 347.99 | 92.50 |
| Nature DQN (200M)** | - | 241.06 | 93.52 |

### 5.3 Analysis on the Role of the Diffusion Factor $\beta$

In this section, we make comparisons between our own EBU algorithms. EBU ($\beta = 1.0$) works the best in the MNIST Maze environment because we use MNIST images for the state representation to allow consecutive states to exhibit little correlation. However, in the Atari domain, consecutive states are often different in a scale of few pixels only. As a consequence, EBU ($\beta = 1.0$) underperforms EBU ($\beta = 0.5$) in most of the Atari games. In order to analyze this phenomenon, we evaluate the $Q$-values learned at the end of each training epoch. We report the test episode score and the corresponding mean $Q$-values of all transitions within the test episode (Figure 5). We notice that the EBU ($\beta = 1.0$) is trained to output highly overestimated $Q$-values compared to its actual return. Since the EBU method performs recursive max operations, EBU outputs higher (possibly overestimated) $Q$-values than Nature DQN. This result indicates that sequentially updating correlated states with

overestimated values may destabilize the learning process. However, this result clearly implies that EBU ($\beta = 0.5$) is relatively free from the overestimation problem.

Next, we investigate the efficacy of using an adaptive diffusion factor. In Figure 6, we present how adaptive EBU adapts its diffusion factor during the course of training in "Breakout". In the early stage of training, the agent barely succeeds in breaking a single brick. With a high $\beta$ close to 1, values can be directly propagated from the rewarded state to the state where the agent has to bounce the ball up. Note that the performance of adaptive EBU follows that of EBU ($\beta = 1.0$) up to about 5M frames. As the training proceeds, the agent encounters more rewards and various trajectories that may cause overestimation. As a consequence, we discover that the agent anneals the diffusion factor to a lower value of 0.5. The trend of how the diffusion factor adapts differs from game to game. Refer to the diffusion factor curves for all 49 games in Appendix C to check how adaptive EBU selects the best diffusion factor.

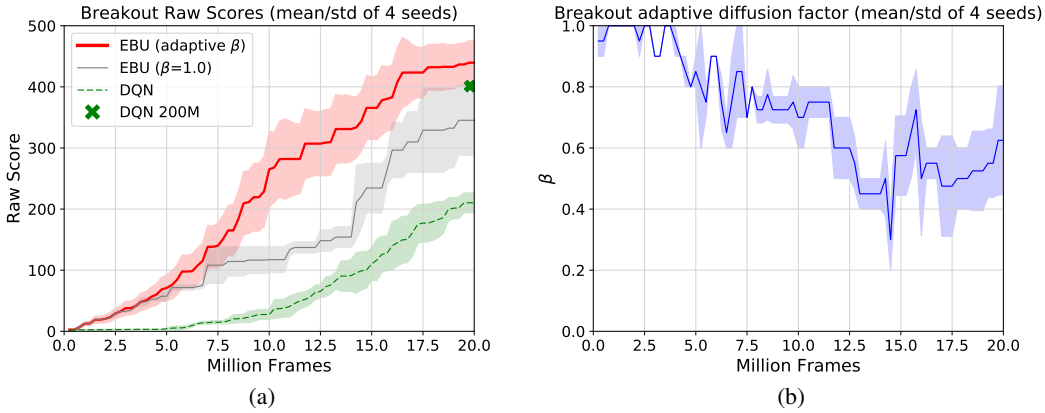

Figure 6: (a) Test scores in "Breakout". Mean and standard deviation from 4 random seeds are plotted. (b) Adaptive diffusion factor of adaptive EBU in "Breakout".

# 6 Conclusion

In this work, we propose Episodic Backward Update, which samples transitions episode by episode, and updates values recursively in a backward manner. Our algorithm achieves fast and stable learning due to its efficient value propagation. We theoretically prove the convergence of our method, and experimentally show that our algorithm outperforms other baselines in many complex domains, requiring only about 10% of samples. Since our work differs from DQN only in terms of the target generation, we hope that we can make further improvements by combining with other successful deep reinforcement learning methods.

**Acknowledgments**

This work was supported by the ICT R&D program of MSIP/IITP. [2016-0-00563, Research on Adaptive Machine Learning Technology Development for Intelligent Autonomous Digital Companion]

## Footnotes

[1]The code is available at `https://github.com/suyoung-lee/Episodic-Backward-Update`

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
