[Supplementary Material · Supp_Episodic Backward Update.pdf]

# Appendix A      Episodic Backward Update with an adaptive diffusion factor

---

**Algorithm 3** Adaptive Episodic Backward Update

---

1: **Initialize**: replay memory $D$ to capacity $N$, $K$ on-line action-value function $Q_1(\cdot; \boldsymbol{\theta_1}), \ldots, Q_K(\cdot; \boldsymbol{\theta_K})$, $K$ target action-value function $\hat{Q}_1(\cdot; \boldsymbol{\theta_1^-}), \ldots, \hat{Q}_K(\cdot; \boldsymbol{\theta_K^-})$, training score recorder $TS = \text{zeros}(K)$, diffusion factors $\beta_1, \ldots, \beta_K$ for each learner network

2: **for** episode = 1 to $M$ **do**

3:      Select $Q_{\text{actor}} = Q_i$ as the actor network for the current episode, where $i = (\text{episode} - 1)\% K + 1$

4:      **for** $t = 1$ to Terminal **do**

5:          With probability $\epsilon$ select a random action $a_t$

6:          Otherwise select $a_t = \text{argmax}_a \, Q_{\text{actor}}(s_t, a)$

7:          Execute action $a_t$, observe reward $r_t$ and next state $s_{t+1}$

8:          Store transition $(s_t, a_t, r_t, s_{t+1})$ in $D$

9:          Add training score for the current learner $TS[i] += r_t$

10:         Sample a random episode $E = \{\boldsymbol{S}, \boldsymbol{A}, \boldsymbol{R}, \boldsymbol{S'}\}$ from $D$, set $T = \text{length}(E)$

11:         **for** $j = 1$ to $K$ (this loop is processed in parallel) **do**

12:             Generate temporary target $Q$-table, $\tilde{Q}_j = \hat{Q}_i\left(\boldsymbol{S'}, \cdot; \boldsymbol{\theta_j^-}\right)$

13:             Initialize target vector $\boldsymbol{y} = \text{zeros}(T)$, $\boldsymbol{y}_T \leftarrow \boldsymbol{R}_T$

14:             **for** $k = T - 1$ to $1$ **do**

15:                 $\tilde{Q}_j\left[\boldsymbol{A}_{k+1}, k\right] \leftarrow \beta_j \boldsymbol{y}_{k+1} + (1 - \beta_j)\tilde{Q}_j\left[\boldsymbol{A}_{k+1}, k\right]$

16:                 $\boldsymbol{y}_k \leftarrow \boldsymbol{R}_k + \gamma \max_a \tilde{Q}_j\left[a, k\right]$

17:             **end for**

18:             Perform a gradient descent step on $\left(\boldsymbol{y} - Q_j\left(\boldsymbol{S}, \boldsymbol{A}; \boldsymbol{\theta_j}\right)\right)^2$ with respect to $\boldsymbol{\theta_j}$

19:         **end for**

20:         Every $C$ steps reset $\hat{Q}_1 = Q_1, \ldots, \hat{Q}_K = Q_K$

21:      **end for**

22:      Every $B$ steps synchronize all learners with the best training score, $b = \text{argmax}_k \, TS[k]$. $Q_1(\cdot; \boldsymbol{\theta_1}) = Q_b(\cdot; \boldsymbol{\theta_b}), \ldots, Q_K(\cdot; \boldsymbol{\theta_K}) = Q_b(\cdot; \boldsymbol{\theta_b})$ and $\hat{Q}_1(\cdot; \boldsymbol{\theta_1}) = \hat{Q}_b(\cdot; \boldsymbol{\theta_b^-}), \ldots, \hat{Q}_K(\cdot; \boldsymbol{\theta_K}) = \hat{Q}_b(\cdot; \boldsymbol{\theta_b^-})$. Reset the training score recorder $TS = \text{zeros}(K)$.

23: **end for**

---

# Appendix B      Raw scores of all 49 games.

Table 1: Raw scores after 10M frames of training. Mean scores from 4 random seeds are reported for adaptive EBU. 8 random seeds are used for all other baselines. We use the results at Nature DQN paper to report the scores at 200M frames. We run their code (https://github.com/deepmind/dqn) to report scores for 10M frames. Due to the use of different random seeds, the result of Nature DQN at 10M frames may be better than that of Nature DQN at 200M frames in some games. Bold texts indicate the best score out of the 5 results trained for 10M frames.

| Training Frames | 10M | | | | | | 20M | 200M |
|---|---|---|---|---|---|---|---|---|
| | EBU($\beta$=0.5) | Adap. EBU | DQN | PER | Retrace($\lambda$) | OT | Adap. EBU | Nature DQN |
| Alien | 708.08 | 894.15 | 690.32 | 1026.96 | 708.29 | **1078.67** | 1225.36 | 3069.00 |
| Amidar | 117.94 | 124.63 | 125.42 | 167.63 | 182.68 | **220.00** | 209.96 | 739.50 |
| Assault | **4109.18** | 3676.95 | 2426.94 | 2720.69 | 2989.05 | 2499.23 | 3943.23 | 3359.00 |
| Asterix | 1898.12 | 2533.27 | **2936.54** | 2218.54 | 1798.54 | 2592.50 | 3221.25 | 6012.00 |
| Asteroids | 1002.17 | **1402.43** | 654.99 | 993.50 | 886.92 | 985.88 | 2378.84 | 1629.00 |
| Atlantis | 61708.75 | 87944.38 | 20666.84 | 35663.83 | **98182.81** | 57520.00 | 141226.00 | 85641.00 |
| Bank heist | 359.62 | **459.42** | 234.70 | 312.96 | 223.50 | 407.42 | 680.43 | 429.70 |
| Battle zone | 20627.73 | 24748.50 | 22468.75 | 20835.74 | **30128.36** | 20400.48 | 30502.53 | 26300.00 |
| Beam rider | 5628.99 | 4785.27 | 3682.92 | 4586.07 | 4093.76 | **5889.54** | 6634.43 | 6846.00 |
| Bowling | 52.02 | **102.89** | 65.23 | 42.74 | 42.62 | 53.45 | 113.75 | 42.40 |
| Boxing | 55.95 | **72.69** | 37.28 | 4.64 | 6.76 | 60.89 | 96.35 | 71.80 |
| Breakout | 174.76 | **265.62** | 28.36 | 164.22 | 171.86 | 75.00 | 443.34 | 401.20 |
| Centipede | 4651.28 | **8389.16** | 6207.30 | 4385.41 | 5986.16 | 5277.79 | 8389.16 | 8309.00 |
| Chopper Command | 1196.67 | 1294.45 | 1168.67 | 1344.24 | 1353.76 | **1615.00** | 1909.23 | 6687.00 |
| Crazy Climber | 65329.63 | **94135.04** | 74410.74 | 53166.47 | 64598.21 | 92972.08 | 103780.15 | 114103.00 |
| Demon Attack | 7924.14 | **8368.16** | 7772.39 | 4446.03 | 6450.84 | 6872.04 | 9099.16 | 9711.00 |
| Double Dunk | -16.19 | **-14.12** | -17.94 | -15.62 | -15.81 | -15.92 | -12.78 | -18.10 |
| Enduro | 415.59 | 326.45 | 516.10 | 308.75 | 208.10 | **615.05** | 410.95 | 301.80 |
| Fishing Derby | -39.13 | **-15.85** | -65.53 | -78.49 | -75.74 | -69.66 | 9.22 | -0.80 |
| Freeway | 19.07 | **23.71** | 16.24 | 9.35 | 15.26 | 14.63 | 34.36 | 30.30 |
| Frostbite | 437.92 | 966.23 | 466.02 | 536.00 | 825.00 | **2452.75** | 1760.15 | 328.30 |
| Gopher | 3318.50 | **3634.67** | 1726.52 | 1833.67 | 3410.75 | 2869.08 | 5611.30 | 8520.00 |
| Gravitar | 294.58 | **450.18** | 193.55 | 319.79 | 272.08 | 263.54 | 611.99 | 306.70 |
| H.E.R.O. | 3089.90 | 3398.55 | 2767.97 | 3052.04 | 3079.43 | **10698.25** | 4308.23 | 19950.00 |
| Ice Hockey | -4.71 | **-2.96** | -4.79 | -7.73 | -6.13 | -5.79 | -2.96 | -1.60 |
| Jamesbond | 391.67 | **519.52** | 183.35 | 421.46 | 436.25 | 325.21 | 1043.66 | 576.70 |
| Kangaroo | 535.83 | 731.13 | 709.88 | **782.50** | 538.33 | 708.33 | 2018.83 | 6740.00 |
| Krull | 7587.24 | 8733.52 | **24109.14** | 6642.58 | 6346.40 | 7468.70 | 10016.72 | 3805.00 |
| Kung-Fu Master | 20578.33 | **26069.68** | 21951.72 | 18212.89 | 18815.83 | 22211.25 | 30387.78 | 23270.00 |
| Montezuma's Revenge | 0.00 | 0.00 | **3.95** | 0.43 | 0.00 | 0.00 | 0.00 | 0.00 |
| Ms. Pacman | 1249.79 | 1652.37 | **1861.80** | 1784.75 | 1310.62 | 1849.00 | 1920.25 | 2311.00 |
| Name This Game | 6960.46 | 7075.53 | **7560.33** | 5757.03 | 6094.08 | 7358.25 | 7565.67 | 7257.00 |
| Pong | 5.53 | **16.49** | -2.68 | 12.83 | 8.65 | 2.60 | 20.23 | 18.90 |
| Private Eye | 471.76 | **3609.96** | 1388.45 | 269.28 | 714.97 | 1277.53 | 7940.27 | 1788.00 |
| Q*Bert | 785.00 | 1074.77 | 2037.21 | 1215.42 | 3192.08 | **3955.10** | 2437.83 | 10596.00 |
| River Raid | 3460.62 | 4268.28 | 3636.72 | 4178.92 | **6005.62** | 4643.62 | 5671.51 | 8316.00 |
| Road Runner | 10086.74 | 15681.49 | 8978.17 | 17137.92 | 9390.83 | **19081.55** | 28286.88 | 18257.00 |
| Robotank | 11.65 | 15.34 | **16.11** | 6.46 | 9.90 | 12.17 | 20.73 | 51.60 |
| Seaquest | 1380.67 | 1926.10 | 762.10 | 1955.67 | 2275.83 | **2710.33** | 5313.43 | 5286.00 |
| Space Invaders | 797.29 | **1058.25** | 755.95 | 762.54 | 783.35 | 869.83 | 1148.21 | 1976.00 |
| Star Gunner | 2737.08 | **3892.51** | 708.66 | 2629.17 | 2856.67 | 1710.83 | 17462.88 | 57997.00 |
| Tennis | -3.41 | -0.96 | **0.00** | -10.32 | -2.50 | -6.37 | -0.93 | -2.50 |
| Time Pilot | 3505.42 | **4567.18** | 3076.98 | 4434.17 | 3651.25 | 4012.50 | 4567.18 | 5947.00 |
| Tutankham | 204.83 | 239.51 | 165.27 | 255.74 | 156.16 | **247.81** | 299.11 | 186.70 |
| Up and Down | 6841.83 | 6754.11 | **9468.04** | 7397.29 | 7574.53 | 6706.83 | 10984.70 | 8456.00 |
| Venture | 105.10 | **194.89** | 96.70 | 60.40 | 50.85 | 106.67 | 242.56 | 380.00 |
| Video Pinball | **84859.24** | 78405.27 | 17803.69 | 55646.66 | 18346.58 | 38528.58 | 84695.96 | 42684.00 |
| Wizard of Wor | 1249.89 | **2030.63** | 529.85 | 1175.24 | 1083.69 | 1177.08 | 4185.40 | 3393.00 |
| Zaxxon | 3221.67 | 3487.38 | 685.84 | **3928.33** | 596.67 | 2467.92 | 6548.52 | 4977.00 |

# Appendix C  Learning curves and corresponding adaptive diffusion factor

Figure 1: Test scores and diffusion factor of Adaptive EBU. We report the mean and the standard deviation from 4 random seeds. We compare the performance of adaptive EBU with the result reported in Nature DQN, trained for 200M frames. The blue curve below each test score plot shows how adaptive EBU adapts its diffusion factor during the course of training.

# Appendix D    Network structure and hyperparameters

## 2D MNIST Maze Environment

Each state is given as a grey scale $28 \times 28$ image. We apply 2 convolutional neural networks (CNNs) and one fully connected layer to get the output $Q$-values for 4 actions: up, down, left and right. The first CNN uses 64 channels with $4 \times 4$ kernels and stride of 3. The next CNN uses 64 channels with $3 \times 3$ kernels and stride of 1. Then the layer is fully connected into a size of 512. Then we fully connect the layer into a size of the action space 4. After each layer, we apply a rectified linear unit.

We train the agent for a total of 200,000 steps. The agent performs $\epsilon$-greedy exploration. $\epsilon$ starts from 1 and is annealed to 0 at 200,000 steps in a quadratic manner: $\epsilon = \frac{1}{(200,000)^2}(\text{step} - 200,000)^2$. We use RMSProp optimizer with a learning rate of 0.001. The online-network is updated every 50 steps, the target network is updated every 2000 steps. The replay memory size is 30000 and we use minibatch size of 350. We use a discount factor $\gamma = 0.9$ and a diffusion factor $\beta = 1.0$. The agent plays the game until it reaches the goal or it stays in the maze for more than 1000 time steps.

## 49 Games of Atari 2600 Domain

### Common specifications for all baselines
Almost all specifications such as hyperparameters and network structures are identical for all baselines. We use exactly the same network structure and hyperparameters of Nature DQN (Mnih et al., 2015). The raw observation is preprocessed into a gray scale image of $84 \times 84$. Then it passes through three convolutional layers: 32 channels with $8 \times 8$ kernels with a stride of 4; 64 channels with $4 \times 4$ kernels with a stride of 2; 64 channels with $3 \times 3$ kernels with a stride of 1. Then it is fully connected into a size of 512. Then it is again fully connected into the size of the action space.

We train baselines for 10M frames each, which is equivalent to 2.5M steps with frameskip of 4. The agent performs $\epsilon$-greedy exploration. $\epsilon$ starts from 1 and is linearly annealed to reach the final value 0.1 at 4M frames of training. We adopt 30 no-op evaluation methods. We use 8 random seeds for 10M frames and 4 random seeds for 20M frames. The network is trained by RMSProp optimizer with a learning rate of 0.00025. At each update (4 agent steps or 16 frames), we update transitions in minibatch with size 32. The replay buffer size is 1 million steps (4M frames). The target network is updated every 10,000 steps. The discount factor is $\gamma = 0.99$.

We divide the training process into 40 epochs (80 epochs for 20M frames) of 250,000 frames each. At the end of each epoch, the agent is tested for 30 episodes with $\epsilon = 0.05$. The agent plays the game until it runs out of lives or time (18,000 frames, 5 minutes in real time).

Below are detailed specifications for each algorithm.

### 1. Episodic Backward Update
We used $\beta = 0.5$ for the version EBU with constant diffusion factor. For adaptive EBU, we used 11 parallel learners ($K = 11$) with diffusion factors 0.0, 0.1, ..., 1.0. We synchronize the learners at every 250,000 frames ($B = 62,500$ steps).

### 2. Prioritized Experience Replay
We use the rank-based DQN version of Prioritized ER and use the hyperparameters chosen by the authors (Schaul et al., 2016): $\alpha = 0.5 \rightarrow 0$ and $\beta = 0$.

### 3. Retrace$(\lambda)$
Just as EBU, we sample a random episode and then generate the Retrace target for the transitions in the sampled episode. We follow the same evaluation process as that of Munos et al., 2016. First, we calculate the trace coefficients from $s = 1$ to $s = T$ (terminal).

$$c_s = \lambda \min\left(1, \frac{\pi(a_s|x_s)}{\mu(a_s|x_s)}\right) \tag{1}$$

Where $\mu$ is the behavior policy of the sampled transition and the evaluation policy $\pi$ is the current policy. Then we generate a loss vector for transitions in the sample episode from $t = T$ to $t = 1$.

$$\Delta Q(x_{t-1}, a_{t-1}) = c_t \lambda \Delta Q(x_t, a_t) + [r(x_{t-1}, a_{t-1}) + \gamma \mathbb{E}_\pi Q(x_t, :) - Q(x_{t-1}, a_{t-1})]. \tag{2}$$

### 4. Optimality Tightening
We use the source code (https://github.com/ShibiHe/Q-Optimality-Tightening), modify the maximum test steps and test score calculation to match the evaluation policy of Nature DQN.

# Appendix E    Supplementary figure: backward update algorithm

Line #7 of Algorithm 2: Sample a random episode $E$.

| $E$ | | | | | | |
|---|---|---|---|---|---|---|
| $S$ | $S_1$ | ... | $S_{T-2}$ | $S_{T-1}$ | $S_T$ | |
| $A$ | $A_1$ | ... | $A_{T-2}$ | $A_{T-1}$ | $A_T$ | |
| $R$ | $R_1$ | ... | $R_{T-2}$ | $R_{T-1}$ | $R_T$ | |
| $S'$ | $S_2$ | ... | $S_{T-1}$ | $S_T$ | $S_{T+1}$ | |

Line # 8~9: Generate a temporary target Q table $\tilde{Q}$ with the next state vector $S'$. Initialize a target vector $y$.
Let there be $n$ possible actions in the environment. $\mathcal{A} = \{a^{(1)}, a^{(2)}, ..., a^{(n)}\}$.
Note that $\hat{Q}$ is the target Q-value and $\hat{Q}(S_{T+1}, :) = 0$.

| $\tilde{Q}$ | $\hat{Q}(S_2, a^{(1)})$ | ... | $\hat{Q}(S_{T-1}, a^{(1)})$ | $\hat{Q}(S_T, a^{(1)})$ | 0 | |
|---|---|---|---|---|---|---|
| | $\hat{Q}(S_2, a^{(2)})$ | ... | $\hat{Q}(S_{T-1}, a^{(2)})$ | $\hat{Q}(S_T, a^{(2)})$ | 0 | $n$ |
| | $\vdots$ | $\vdots$ | $\vdots$ | $\vdots$ | $\vdots$ | |
| | $\hat{Q}(S_2, a^{(n)})$ | ... | $\hat{Q}(S_{T-1}, a^{(n)})$ | $\hat{Q}(S_T, a^{(n)})$ | 0 | |
| $y$ | 0 | ... | 0 | 0 | $R_T$ | |

Line # 10~12, first iteration (k = T-1): Update $\tilde{Q}$ and $y$. Let the T-th action in the replay memory be $A_T = a^{(2)}$.
  ① line # 15: update $\tilde{Q}[A_{k+1}, k] = \tilde{Q}[A_T, T-1] = \tilde{Q}[a^{(2)}, T-1] \leftarrow \beta\, y_T + (1-\beta)\hat{Q}(S_T, a^{(2)})$
  ② line # 16: update $y_k = y_{T-1} \leftarrow R_{T-1} + \gamma \max \tilde{Q}[:, T-1]$

| | index | 1 | ... | T-2 | T-1 | T | |
|---|---|---|---|---|---|---|---|
| $\tilde{Q}$ | $a^{(1)}$ | $\hat{Q}(S_2, a^{(1)})$ | ... | $\hat{Q}(S_{T-1}, a^{(1)})$ | $\hat{Q}(S_T, a^{(1)})$ | 0 | |
| | $a^{(2)}$ | $\hat{Q}(S_2, a^{(2)})$ | ... | $\hat{Q}(S_{T-1}, a^{(2)})$ | $\beta\, y_T + (1-\beta)\hat{Q}(S_T, a^{(2)})$ | 0 | $n$ |
| | $\vdots$ | $\vdots$ | $\vdots$ | $\vdots$ | $\vdots$ | $\vdots$ | |
| | $a^{(n)}$ | $\hat{Q}(S_2, a^{(n)})$ | ... | $\hat{Q}(S_{T-1}, a^{(n)})$ | $\hat{Q}(S_T, a^{(n)})$ | 0 | |
| $y$ | | 0 | ... | 0 | $R_{T-1} + \gamma \max \tilde{Q}[:, T-1]$ | $R_T$ | |

Line # 10~12, second iteration (k = T-2): Update $\tilde{Q}$ and $y$. Let the (T-1)-th action in the replay memory be $A_{T-1} = a^{(1)}$.
  ① line # 15: update $\tilde{Q}[A_{k+1}, k] = \tilde{Q}[A_{T-1}, T-2] = \tilde{Q}[a^{(1)}, T-2] \leftarrow \beta\, y_{T-1} + (1-\beta)\hat{Q}(S_{T-1}, a^{(1)})$
  ② line # 16: update $y_k = y_{T-2} \leftarrow R_{T-2} + \gamma \max \tilde{Q}[:, T-2]$

| | index | 1 | ... | T-2 | T-1 | T | |
|---|---|---|---|---|---|---|---|
| $\tilde{Q}$ | $a^{(1)}$ | $\hat{Q}(S_2, a^{(1)})$ | ... | $\beta\, y_{T-1} + (1-\beta)\hat{Q}(S_{T-1}, a^{(1)})$ | $\hat{Q}(S_T, a^{(1)})$ | 0 | |
| | $a^{(2)}$ | $\hat{Q}(S_2, a^{(2)})$ | ... | $\hat{Q}(S_{T-1}, a^{(2)})$ | $\beta\, y_T + (1-\beta)\hat{Q}(S_T, a^{(2)})$ | 0 | $n$ |
| | $\vdots$ | $\vdots$ | $\vdots$ | $\vdots$ | $\vdots$ | $\vdots$ | |
| | $a^{(n)}$ | $\hat{Q}(S_2, a^{(n)})$ | ... | $\hat{Q}(S_{T-1}, a^{(n)})$ | $\hat{Q}(S_T, a^{(n)})$ | 0 | |
| $y$ | | 0 | ... | $R_{T-2} + \gamma \max \tilde{Q}[:, T-2]$ | $R_{T-1} + \gamma \max \tilde{Q}[:, T-1]$ | $R_T$ | |

Repeat this update until k =1.

Figure 2: Target generation process from the sampled episode E

## Appendix F    Comparison to other multi-step methods.

Figure 3: A motivating example where $Q(\lambda)$ underperforms Episodic Backward Update. **Left:** A simple navigation domain with 3 possible episodes. $s_1$ is the initial state. States with ' signs are the terminal states. **Right:** An extended example with $n$ possible episodes.

Imagine a toy navigation environment as in Figure 3, left. Assume that an agent has experienced all possible trajectories: $(s_1 \to s_1^{'})$; $(s_1 \to s_2 \to s_2^{'})$ and $(s_1 \to s_2 \to s_3^{'})$. Let the discount factor $\gamma$ be 1. Then optimal policy is $(s_1 \to s_2 \to s_3^{'})$. With a slight abuse of notation let $Q(s_i, s_j)$ denote the value of the action that leads to the state $s_j$ from the state $s_i$. We will show that $Q(\lambda)$ and $Q^*(\lambda)$ methods underperform Episodic Backward Update in such examples with many suboptimal branching paths.

$Q(\lambda)$ method cuts trace of the path when the path does not follow greedy actions given the current $Q$-value. For example, assume a $Q(\lambda)$ agent has updated the value $Q(s_1, s_1^{'})$ at first. When the agent tries to update the values of the episode $(s_1 \to s_2 \to s_3^{'})$, the greedy policy of the state $s_1$ heads to $s_1^{'}$. Therefore the trace of the optimal path is cut and the reward signal $r_3$ is not passed to $Q(s_1, s_2)$. This problem becomes more severe if the number of suboptimal branches increases as illustrated in Figure 3, right. Other variants of $Q(\lambda)$ algorithm that cut traces, such as Retrace($\lambda$), have the same problem. EBU does not suffer from this issue, because EBU does not cut the trace, but performs max operations at every branch to propagate the maximum value.

$Q^*(\lambda)$ is free from the issues mentioned above since it does not cut traces. However, to guarantee convergence to the optimal value function, it requires the parameter $\lambda$ to be less than $\frac{1-\gamma}{2\gamma}$. In convention, the discount factor $\gamma \approx 1$. For a small value of $\lambda$ that satisfies the constraint, the update of distant returns becomes nearly negligible. However, EBU does not have any constraint of the diffusion factor $\beta$ to guarantee convergence.

## Appendix G    Theoretical guarantees

Now, we will prove that the episodic backward update algorithm converges to the true action-value function $Q^*$ in the case of finite and deterministic environment.

**Definition 1.** *(Deterministic MDP)*

$M = (\mathcal{S}, \mathcal{A}, P, R)$ *is a* ***deterministic MDP*** *if* $\exists g : \mathcal{S} \times \mathcal{A} \to \mathcal{S}$ *s.t.*

$$P(s'|s,a) = \begin{cases} 1 & \text{if } s' = g(s,a) \\ 0 & \text{else} \end{cases} \quad \forall (s,a,s') \in \mathcal{S} \times \mathcal{A} \times \mathcal{S},$$

In the episodic backward update algorithm, a single (state, action) pair can be updated through multiple episodes, where the evaluated targets of each episode can be different from each other. Therefore, unlike the bellman operator, episodic backward operator depends on the exploration policy for the MDP. Therefore, instead of expressing different policies in each state, we define a schedule to represent the frequency of every distinct episode (which terminates or continues indefinitely) starting from the target (state, action) pair.

**Definition 2.** *(Schedule)*

*Assume a MDP* $M = (\mathcal{S}, \mathcal{A}, P, R)$ *, where $R$ is a bounded function. Then, for each state $(s,a) \in \mathcal{S} \times \mathcal{A}$ and $j \in [1, \infty]$, we define $j$-**length path set** $p_{s,a}(j)$ and **path set** $p(s,a)$ for $(s,a)$ as*

$$p_{s,a}(j) = \left\{ (s_i, a_i)_{i=0}^{j} | (s_0, a_0) = (s,a), P(s_{i+1}|s_i, a_i) > 0 \quad \forall i \in [0, j-1], s_j \quad is \quad terminal \right\}.$$

*and* $p_{s,a} = \cup_{j=1}^{\infty} p_{s,a}(j)$.

*Also, we define a **schedule set** $\lambda_{s,a}$ for (state action) pair $(s,a)$ as*

$$\lambda_{s,a} = \left\{ (\lambda_i)_{i=1}^{|p_{s,a}|} | \sum_{i=1}^{|p_{s,a}|} \lambda_i = 1, \lambda_i > 0 \quad \forall i \in [1, |p_{s,a}|] \right\}.$$

*Finally, to express the varying schedule in time at the RL scenario, we define a **time schedule set** $\lambda$ for MDP $M$ as*

$$\lambda = \left\{ \{\lambda_{s,a}(t)\}_{(s,a) \in \mathcal{S} \times \mathcal{A}, t=1}^{\infty} | \lambda_{s,a}(t) \in \lambda_{s,a}, \forall (s,a) \in \mathcal{S} \times \mathcal{A}, t \in [1, \infty] \right\}.$$

Since no element of the path can be the prefix of the others, the path set corresponds to the enumeration of all possible episodes starting from each (state, action) pair. Therefore, if we utilize multiple episodes from any given policy, we can see the empirical frequency for each path in the path set belongs to the schedule set. Finally, since the exploration policy can vary across time, we can group independent schedules into the time schedule set.

For a given time schedule and MDP, now we define the episodic backward operator.

**Definition 3.** *(Episodic backward operator)*

*For an MDP* $M = (\mathcal{S}, \mathcal{A}, P, R)$, *and a time schedule* $\{\lambda_{s,a}(t)\}_{t=1,(s,a) \in \mathcal{S} \times \mathcal{A}}^{\infty} \in \lambda$.

*Then, the **episodic backward operator** $H_t^{\beta}$ is defined as*

$$(H_t^{\beta} Q)(s,a) \tag{3}$$

$$= \mathbb{E}_{s' \in \mathcal{S}, P(s'|s,a)} \left[ r(s,a,s') + \gamma \sum_{i=1}^{|p_{s,a}|} (\lambda_{(s,a)}(t))_i \mathbb{1}(s_{i1} = s') \left[ \max_{1 \leq j \leq |(p_{s,a})_i|} T_{(p_{s,a})_i}^{\beta, Q}(j) \right] \right].$$

$$T_{(p_{s,a})_i}^{\beta, Q}(j) \tag{4}$$

$$= \sum_{k=1}^{j-1} \beta^{k-1} \gamma^{k-1} \left\{ \beta r(s_{ik}, a_{ik}, s_{i(k+1)}) + (1-\beta) Q(s_{ik}, a_{ik}) \right\} + \beta^{j-1} \gamma^{j-1} \max_{a \neq a_j} Q(s_{ij}, a_{ij}).$$

*Where $(p_{s,a})_i$ is the $i$-th path of the path set, and $(s_{ij}, a_{ij})$ corresponds to the $j$-th (state, action) pair of the $i$-th path.*

Episodic backward operator consists of two parts. First, given the path that initiates from the target (state, action) pair, the function $T^{\beta,Q}_{(p_{s,a})_i}$ computes the maximum return of the path via backward update. Then, the return is averaged by every path in the path set. Now, if the MDP $M$ is deterministic, we can prove that the episodic backward operator is a contraction in the sup-norm, and the fixed point of the episodic backward operator is the optimal action-value function of the MDP regardless of the time schedule.

**Theorem 2.** *(Contraction of the episodic backward operator and the fixed point)*

*Suppose $M = (\mathcal{S}, \mathcal{A}, P, R)$ is a deterministic MDP. Then, for any time schedule $\{\lambda_{s,a}(t)\}^{\infty}_{t=1,(s,a)\in\mathcal{S}\times\mathcal{A}} \in \lambda$, $H^{\beta}_t$ is a contraction in the sup-norm for any t, i.e*

$$\|(H^{\beta}_t Q_1) - (H^{\beta}_t Q_2)\|_{\infty} \leq \gamma \|Q_1 - Q_2\|_{\infty}. \tag{5}$$

*Furthermore, for any time schedule $\{\lambda_{s,a}(t)\}^{\infty}_{t=1,(s,a)\in\mathcal{S}\times\mathcal{A}} \in \lambda$, the fixed point of $H^{\beta}_t$ is the optimal $Q$ function $Q^*$.*

*Proof.* First, we prove $T^{\beta,Q}_{(p_{s,a})_i}(j)$ is a contraction in the sup-norm for all $j$.

Since $M$ is a deterministic MDP, we can reduce the return as

$$T^{\beta,Q}_{(p_{s,a})_i}(j) = \left( \sum_{k=1}^{j-1} \beta^{k-1}\gamma^{k-1} \{\beta r(s_{ik}, a_{ik}) + (1-\beta)Q(s_{ik}, a_{ik})\} + \beta^{j-1}\gamma^{j-1} \max_{a \neq a_j} Q(s_{ij}, a_{ij}) \right). \tag{6}$$

$$\begin{aligned}
\|T^{\beta,Q_1}_{(p_{s,a})_i}(j) - T^{\beta,Q_2}_{(p_{s,a})_i}(j)\|_{\infty} &\leq \left\{ (1-\beta) \sum_{k=1}^{j-1} \beta^{k-1}\gamma^{k-1} + \beta^{j-1}\gamma^{j-1} \right\} \|Q_1 - Q_2\|_{\infty} \\
&= \left\{ \frac{(1-\beta)(1-(\beta\gamma)^{j-1})}{1-\beta\gamma} + \beta^{j-1}\gamma^{j-1} \right\} \|Q_1 - Q_2\|_{\infty} \\
&= \frac{1 - \beta + \beta^j \gamma^{j-1} - \beta^j \gamma^j}{1 - \beta\gamma} \|Q_1 - Q_2\|_{\infty} \\
&= \left\{ 1 + (1-\gamma) \frac{\beta^j \gamma^{j-1} - \beta}{1-\beta\gamma} \right\} \|Q_1 - Q_2\|_{\infty} \\
&\leq \|Q_1 - Q_2\|_{\infty} \quad (\because \beta \in [0,1], \gamma \in [0,1)).
\end{aligned} \tag{7}$$

Also, at the deterministic MDP, the episodic backward operator can be reduced to

$$(H^{\beta}_t Q)(s, a) = r(s, a) + \gamma \sum_{i=1}^{|p_{s,a}|} (\lambda_{(s,a)})_i(t) \left[ \max_{1 \leq j \leq |(p_{s,a})_i|} T^{\beta,Q}_{(p_{s,a})_i}(j) \right]. \tag{8}$$

Therefore, we can finally conclude that

$$\|(H_t^\beta Q_1) - (H_t^\beta Q_2)\|_\infty$$

$$= \max_{s,a} \left| H_t^\beta Q_1(s,a) - H_t^\beta Q_2(s,a) \right|$$

$$\leq \gamma \max_{s,a} \left[ \sum_{i=1}^{|p_{s,a}|} (\lambda_{(s,a)}(t))_i \left| \left\{ \max_{1 \leq j \leq |(p_{s,a})_i|} T_{(p_{s,a})_i}^{\beta,Q_1}(j) \right\} - \left\{ \max_{1 \leq j \leq |(p_{s,a})_i|} T_{(p_{s,a})_i}^{\beta,Q_2}(j) \right\} \right| \right]$$

$$\leq \gamma \max_{s,a} \left[ \sum_{i=1}^{|p_{s,a}|} (\lambda_{(s,a)}(t))_i \max_{1 \leq j \leq |(p_{s,a})_i|} \left\{ \left| T_{(p_{s,a})_i}^{\beta,Q_1}(j) - T_{(p_{s,a})_i}^{\beta,Q_2}(j) \right| \right\} \right]$$

$$\leq \gamma \max_{s,a} \left[ \sum_{i=1}^{|p_{s,a}|} (\lambda_{(s,a)}(t))_i \|Q_1 - Q_2\|_\infty \right]$$

$$= \gamma \max_{s,a} \left[ \|Q_1 - Q_2\|_\infty \right]$$

$$= \gamma \|Q_1 - Q_2\|_\infty. \tag{9}$$

Therefore, we have proved that the episodic backward operator is a contraction independent of the schedule. Finally, we prove that the distinct episodic backward operators in terms of schedule have the same fixed point, $Q^*$. A sufficient condition to prove this is given by

$$\left[ \max_{1 \leq j \leq |(p_{s,a})_i|} T_{(p_{s,a})_i}^{\beta,Q^*}(j) \right] = \frac{Q^*(s,a) - r(s,a)}{\gamma} \ \forall 1 \leq i \leq |p_{s,a}|.$$

We will prove this by contradiction. Assume $\exists i$ s.t. $\left[ \max_{1 \leq j \leq |(p_{s,a})_i|} T_{(p_{s,a})_i}^{\beta,Q^*}(j) \right] \neq \frac{Q^*(s,a) - r(s,a)}{\gamma}$.

First, by the definition of $Q^*$ fuction, we can bound $Q^*(s_{ik}, a_{ik})$ and $Q^*(s_{ik}, :)$ for every $k \geq 1$ as follows.

$$Q^*(s_{ik}, a) \leq \gamma^{-k} Q^*(s,a) - \sum_{m=0}^{k-1} \gamma^{m-k} r(s_{im}, a_{im}). \tag{10}$$

Note that the equality holds if and only if the path $(s_i, a_i)_{i=0}^{k-1}$ is the optimal path among the ones that start from $(s_0, a_0)$. Therefore, $\forall 1 \leq j \leq |(p_{s,a})_i|$, we can bound $T_{(p_{s,a})_i}^{\beta,Q^*}(j)$.

$$T^{\beta,Q}_{(p_{s,a})_i}(j)$$

$$= \sum_{k=1}^{j-1} \beta^{k-1}\gamma^{k-1}\left\{\beta r(s_{ik},a_{ik}) + (1-\beta)Q(s_{ik},a_{ik})\right\} + \beta^{j-1}\gamma^{j-1}\max_{a\neq a_j} Q(s_{ij},a_{ij})$$

$$\leq \left\{\left(\sum_{k=1}^{j-1}(1-\beta)\beta^{k-1}\right) + \beta^{j-1}\right\}\gamma^{-1}Q^*(s,a)$$
$$+ \sum_{k=1}^{j-1}\left\{\beta^{k-1}\gamma^{k-1}\left(\beta r(s_{ik},a_{ik}) - \sum_{m=0}^{k-1}(1-\beta)\gamma^{m-k}r(s_{im},a_{im})\right)\right\}$$
$$- \sum_{m=0}^{j-1}\beta^{j-1}\gamma^{j-1}\gamma^{m-j}r(s_{im},a_{im})$$

$$= \gamma^{-1}Q^*(s,a) + \sum_{k=1}^{j-1}\beta^k\gamma^{k-1}r(s_{ik},a_{ik})$$
$$- \sum_{m=0}^{j-2}\left\{\sum_{k=m+1}^{j-1}(1-\beta)\beta^{k-1}\gamma^{m-1}r(s_{im},a_{im})\right\} - \sum_{m=0}^{j-1}\beta^{j-1}\gamma^{m-1}r(s_{im},a_{im})$$

$$= \gamma^{-1}Q^*(s,a) + \sum_{m=1}^{j-1}\beta^m\gamma^{m-1}r(s_{im},a_{im})$$
$$- \sum_{m=0}^{j-2}(\beta^m - \beta^{j-1})\gamma^{m-1}r(s_{im},a_{im}) - \sum_{m=0}^{j-1}\beta^{j-1}\gamma^{m-1}r(s_{im},a_{im})$$

$$= \gamma^{-1}Q^*(s,a) - \gamma^{-1}r(s_{i0},a_{i0}) = \frac{Q^*(s,a) - r(s,a)}{\gamma}.$$

(11)

Since this occurs for any arbitrary path, the only remaining case is when

$\exists i$ s.t. $\left[\max_{1\leq j\leq|(p_{s,a})_i|} T^{\beta,Q^*}_{(p_{s,a})_i}(j)\right] < \frac{Q^*(s,a)-r(s,a)}{\gamma}$.

Now, let's turn our attention to the path $s_0, s_1, s_2, ...., s_{|(p_{s,a})_i)|}$. Let's first prove the contradiction when the length of the contradictory path is finite. If $Q^*(s_{i1},a_{i1}) < \gamma^{-1}(Q^*(s,a)-r(s,a))$, then by the Bellman equation, there exists an action $a \neq a_{i1}$ s.t. $Q^*(s_{i1},a) = \gamma^{-1}(Q^*(s,a)-r(s,a))$. Then, we can find that $T^{\beta,Q^*}_{(p_{s,a})_1}(1) = \gamma^{-1}(Q^*(s,a)-r(s,a))$. It contradicts the assumption, therefore $a_{i1}$ should be the optimal action in $s_{i1}$.

Repeating the procedure, we conclude that $a_{i1}, a_{i2}, ..., a_{|(p_{s,a})_i)|-1}$ are optimal with respect to their corresponding states.

Finally, $T^{\beta,Q^*}_{(p_{s,a})_1}(|(p_{s,a})_i)|) = \gamma^{-1}(Q^*(s,a)-r(s,a))$ since all the actions satisfy the optimality condition of the inequality in equation 7. Therefore, it contradicts the assumption.

In the case of an infinite path, we will prove that for any $\epsilon > 0$, there is no path that satisfies $\frac{Q^*(s,a)-r(s,a)}{\gamma} - \left[\max_{1\leq j\leq|(p_{s,a})_i|} T^{\beta,Q^*}_{(p_{s,a})_i}(j)\right] = \epsilon$.

Since the reward function is bounded, we can define $r_{\max}$ as the supremum norm of the reward function. Define $q_{\max} = \max_{s,a}|Q(s,a)|$ and $R_{\max} = \max\{r_{\max}, q_{\max}\}$. We can assume $R_{\max} > 0$. Then, let's set $n_\epsilon = \lceil \log_\gamma \frac{\epsilon(1-\gamma)}{R_{\max}} \rceil + 1$. Since $\gamma \in [0,1)$, $R_{\max}\frac{\gamma^{n_\epsilon}}{1-\gamma} < \epsilon$. Therefore, by applying the procedure on the finite path case for $1 \le j \le n_\epsilon$, we can conclude that the assumption leads to a contradiction. Since the previous $n_\epsilon$ trajectories are optimal, the rest trajectories can only generate a return less than $\epsilon$.

Finally, we proved that $\left[ \max_{1 \le j \le |(p_{s,a})_i|} T^{\beta,Q^*}_{(p_{s,a})_i}(j) \right] = \frac{Q^*(s,a) - r(s,a)}{\gamma}$ $\forall 1 \le i \le |p_{s,a}|$ and therefore, every episodic backward operator has $Q^*$ as the fixed point. $\qquad\square$

Finally, we will show that the online episodic backward update algorithm converges to the optimal $Q$ function $Q^*$.

**Restatement of Theorem 1.** *Given a finite, deterministic, and tabular MDP $M = (\mathcal{S}, \mathcal{A}, P, R)$, the episodic backward update algorithm, given by the update rule*

$Q_{t+1}(s_t, a_t)$

$= (1 - \alpha_t)Q_t(s_t, a_t) + \alpha_t \left[ r(s_t, a_t) + \gamma \sum_{i=1}^{|p_{s_t,a_t}|}(\lambda_{(s_t,a_t)})_i(t) \left[ \max_{1 \le j \le |(p_{s_t,a_t})_i|} T^{\beta,Q}_{(p_{s_t,a_t})_i}(j) \right] \right]$

*converges to the optimal Q-function w.p. 1 as long as*

- *The step size satisfies the Robbins-Monro condition;*

- *The sample trajectories are finite in lengths $l$: $\mathbb{E}[l] < \infty$;*

- *Every (state, action) pair is visited infinitely often.*

For the proof of Theorem 1, we follow the proof of [Melo, 2001](#).

**Lemma 1.** *The random process $\Delta_t$ taking values in $\mathbb{R}^n$ and defined as*

$\Delta_{t+1}(x) = (1 - \alpha_t(x))\Delta_t(x) + \alpha_t(x)F_t(x)$

*converges to zero w.p. 1 under the following assumptions:*

- $0 \le \alpha_t \le 1, \sum_t \alpha_t(x) = \infty$ *and* $\sum_t \alpha_t^2(x) < \infty$;

- $\|\mathbb{E}[F_t(x)|\mathcal{F}_t]\|_W \le \gamma\|\Delta_t\|_W$, *with* $\gamma < 1$;

- $\mathbf{var}[F_t(x)|\mathcal{F}_t] \le C\left(1 + \|\Delta_t\|_W^2\right)$, *for* $C > 0$.

By Lemma 1, we can prove that the online episodic backward update algorithm converges to the optimal $Q^*$.

*Proof.* First, by assumption, the first condition of Lemma 1 is satisfied. Also, we can see that by substituting $\Delta_t(s,a) = Q_t(s,a) - Q^*(s,a)$, and $F_t(s,a) = r(s,a) + \gamma \sum_{i=1}^{|p_{s,a}|}(\lambda_{(s,a)})_i(t)\left[\max_{1 \le j \le |(p_{s,a})_i|} T^{\beta,Q}_{(p_{s,a})_i}(j)\right] - Q^*(s,a)$. $\|\mathbb{E}[F_t(s,a)|\mathcal{F}_t]\|_\infty = \|(H_t^\beta Q_t)(s,a) - (H_t^\beta Q^*)(s,a)\|_\infty \le \gamma\|\Delta_t\|_\infty$, where the inequality holds due to the contraction of the episodic backward operator.

Then, $\mathbf{var}[F_t(x)|\mathcal{F}_t] = \mathbf{var}\left[ r(s,a) + \gamma \sum_{i=1}^{|p_{s,a}|}(\lambda_{(s,a)})_i(t)\left[\max_{1 \le j \le |(p_{s,a})_i|} T^{\beta,Q}_{(p_{s,a})_i}(j)\right] \middle| \mathcal{F}_t \right]$.

Since the reward function is bounded, the third condition also holds as well. Finally, by Lemma 1, $Q_t$ converges to $Q^*$.

$\qquad\square$

Although the episodic backward operator can accommodate infinite paths, the operator can be practical when the maximum length of the episode is finite. This assumption holds for many RL domains, such as the ALE.