[Reviews · NeurIPS 2019]

Reviewer 1



The paper proposes to use episodic backwards updates to improve data efficiency in RL tasks, furthermore they introduce a soft relaxation of this in order to combat the overestimation that typically comes from using backwards updates when using Neural Network models. Overall the paper is very clearly written. My main concerns with the paper are in the experimental details as well as in the literature review, also when taking into account the existing literature the novelty of the work is quite limited. The idea of using backwards updates is quite old and goes back to at least the 1993 paper "Prioritized Sweeping" by Moore and Atkeson, which in fact demonstrates a method that is very similar to what the authors propose and which the authors fail to cite. Furthermore recently there were quite a few papers operating in a similar space of ideas using a backward view in ways similar to the authors, e.g.: Fast deep reinforcement learning using online adjustments from the past, https://arxiv.org/abs/1810.08163 RUDDER: Return Decomposition for Delayed Rewards, https://arxiv.org/abs/1806.07857 Is prioritized sweeping the better episodic control?, https://arxiv.org/abs/1711.06677 Efficient Model-Based Deep Reinforcement Learning with Variational State Tabulation, https://arxiv.org/abs/1802.04325 To name a few, these works should be included in the related work section and the precise differences/advantages relative to what the authors propose discussed. Their experimental section is overall well written but there are a few points of concern: It is not clear how the authors picked their hyperparameters, were those selected to give good performance after 20M frames? Did the authors apply the same hyperparameter selection criteria to the baselines? In a recent paper (https://arxiv.org/abs/1906.05243) it has been shown that dqn can be made vastly more data efficient at 10M frames if hyperparameters are optimised to do so. So it's very important in a work focusing on data efficiency to select hyperparameters of baselines and the proposed algorithm according to the same criteria, it is not clear to me that the authors did so. For example from the supplementary material the authors seem to be training their model on a batch every agent step (or 4 frames) while anything DQN based like their prioritized replay baselines will only do so every 4 transitions (or 16 frames), which is in fact one of the changes the authors of https://arxiv.org/abs/1906.05243 advocated to make dqn more data efficient. Also the training procedure for adaptive diffusion parameter is confusing, doesn't the procedure the authors advocate make the training K times less data efficient? Otherwise I believe the paper to be well written and interesting, so my main concern would be with a lack of novelty given recent research as well as some lacking details regarding the experiments. Post Rebuttal: Overall the authors adressed my main concerns. However I found the response of the authors overly combative, especially since one of my main concerns with the paper was a factually incorrect statement regarding the frequency of updates to the policy. I would encourage the authors to take more care in the future when formulating responses to reviewers. Nevertheless the authors adressed my main concern and accordingly I changed my score to 6.

Reviewer 2



The paper is well written and organized, the appendix is extensive. A new heuristics, called episodic backward update (EBU) is presented. The heuristics aims at increasing data-efficiency in sparse reward environments. To mitigate the instability, that this less random approach introduces, a diffusion parameter is proposed together with EBU. A method for adaptive selection of the diffusion parameter is presented. This is replacing a more general approach by a heuristic, that even needs another parameter to be robust. This is not a good idea in general, but in the domain of sparse rewards the paper shows, that the heuristic has its advantages. So it might well be a method that will be used in these domains. The plots in Figure 5 are very interesting, showing some problems of the methods (which I believe are common in Q-function based methods, but not so often discussed). A few suggestions/comments: The comparison of "39 days" with "a couple of hours" is quite unfair, as in the former case, vision has to be learned as well. In subsection 3.1 the Q is not bold, you could use \boldmath{Q} The font of Figure 2, 4, 5, and 6 is very small, practically unreadable.

Reviewer 3



General comments: the paper presents a theoretically sound algorithm which also boosts practical performance of deep RL algorithms. The motivation of the original n-step Q-learning update is to ensure that reward signals can propagate effectively throughout a sampled trajectory, which is aligned with the method proposed in the paper. To address the issue with correlated states and overestimation, the author proposes a diffusion scheme that mediates the learning. Detailed question: 1. The method seems simple yet effective. I feel like a major issue with the implementation is that sampling a whole episode is more complicated than sampling a fixed length trajectory. Does this occur in practice? 2. In Section 3.3, the author introduces an adaptive beta scheme, which could potentially allow for doing without an additional hyperparameter search. However, my understanding is that in practice, the algorithm trains K networks with varying beta in parallel and picks the best one to proceed. This feels like carrying out hyperparameter search on the fly instead of being 'adaptive'? 3. If the n-step Q-learning has a large n (such that n > T, T the episodic horizon), will this correspond exactly to the episodic backward update?

[Author Response · NeurIPS 2019]

**Reviewer #1**

1. (Hyperparameters) In the Atari experiment section 5.2, we have clearly stated as follows.

*"We train EBU and baselines for 10M frames (additional 20M frames for adaptive EBU) on 49 Atari games with the same network structure, hyperparameters, and evaluation methods used in Nature DQN"*

Just as stated, to train all the baselines (Nature DQN, PER, Retrace and OT) and our EBU, we used the **same hyperparameters, network structures and evaluation methods that are already defined in Nature DQN paper**. The common specifications in Appendix D are just detailed descriptions of each hyperparameter used in Nature DQN paper that we applied to all the baselines and our method for the experiment. We **did not select any of the parameters manually** to give a good score at 20M frames. We implemented our algorithm based on the code of Nature DQN **by only modifying the target generation process** and episodic sampling process without modifying a single hyperparameter of the original DQN code. Therefore, we are certain that we have provided a fair comparison.

We apologize for the source of confusion about the update period in Appendix D. We meant *"At each **update step (4 agent steps or 16 frames)**, we update transitions in minibatch with size 32."* We will correct this to prevent confusion.

2. (Literature Review) Our major focus of this research was to improve the sample efficiency of **model-free, value-based deep** reinforcement learning by making a change as simple but effective as possible – that is to modify the target generation method only. There have been many recent related works including the ones the Reviewer 1 cited. Unfortunately, we could not provide detailed differences/advances of them all in the limited amount of manuscript. We decided to include the most related works such as PER, Retrace, OT that improved the sample efficiency of DQN by **only modifying the target generation process** and do not modify the network or add an additional memory structure.

Many of the recent reinforcement learning methods require changes in the network structures or require additional memory structures (Ephemeral Value Adjustments, RUDDER). Some works investigate algorithms in tabular environments or require tabularization (Is prioritized sweeping the better episodic control?, Efficient Model-Based Deep reinforcement learning with variational state tabulation). These works are orthogonal to EBU, which is a model-free, value-based, deep reinforcement learning algorithm that only modifies the target generation process from the original DQN.

The idea of the backward update is not novel and we have stated in section 3.1 that the tabular backward update (Algorithm 1) is a special case of Lin's method (1992). And we also stated that our main contribution is that we successfully applied the backward update idea in the **deep reinforcement learning domain**, which often fails due to the state correlation. The idea of the backward update may be similar to prioritized sweeping (1993) but prioritized sweeping requires a queue to seek for all predecessor states for value updates. Therefore, it is often inapplicable in the deep learning domain or it requires a state tabularization. However, we agree that some of the recent works share the backward update idea and we will try to include this literature review in the manuscript or at least in the appendix.

3. (Data efficiency of the adaptive scheme) Following the convention, we defined the number of **agent-environment interaction** as the metric for data-efficiency. The training process of the adaptive scheme is described in Appendix A. All the K networks are trained using the same sample episode at the same time. All the networks share the same replay memory. Only one of the K networks is selected at every episode to output a policy and to fill the shared replay. Therefore the adaptive method may be K times computationally inefficient to train but achieves the same data efficiency as the constant diffusion factor method.

**Reviewer #2**

1. We agree that the comparison of "39 days" and "a couple of hours" is unfair. We will remove the direct comparison.

2. We apologize for the readability issues, we will certainly modify the figures and text to improve readability.

**Reviewer #3**

1. To sample an episode, we sample one of the terminal states in the replay memory. After sampling a terminal state, we used the episode that includes the terminal state to generate a temporary Q-table and update the values. Therefore, it was not more complicated than sampling a fixed-length trajectory.

2. The goal of the adaptive method is to improve the constant method without harming the data efficiency. We agree that the word 'adaptive' may not be the best description of the method. We will try to find a better description.

3. Even if the n-step Q-learning has a larger n, it does not correspond to the episodic backward update. N-step Q-learning uses the sum of discounted rewards plus the n-step bootstrapped value at the end. However episodic backward update takes the discounted sum of maximum values from backward, so EBU may propagate higher values faster.

[Meta-Review · NeurIPS 2019]

All reviewers recommend accepting the paper. The authors response did address most of the reviewers' concerns. While the AC recommends accepting the paper, the AC encourages the authors to consider the comments of reviewer 1. Specifically, regarding the literature review as well as the hyper-parameter selection in the experimental section. Only changing the backup mechanism keeping all other hyper parameters fixed as in the Nature DQN model is indeed a good experimental setup. However, the optimal operation mode for different models might be different (even when sharing architectures and training protocols): for instance we could 'afford' a larger learning rate if we have a better back-up mechanism. Furthermore it would be informative to include experiments for longer than 10M frames (at least on some key games). In any case, the paper is novel and it is certainly valuable from a practical perspective to successfully implement the backward update idea in the deep reinforcement learning domain (with the current empirical evaluation). Theoretical convergence analysis of the algorithm is also valuable.